 **eLIFE**

# Bidirectional helical motility of cytoplasmic dynein around microtubules

**Sinan Can[1], Mark A Dewitt[2], Ahmet Yildiz[1,2]\***

[1]Department of Physics, University of California, Berkeley, Berkeley, United States; [2]Biophysics Graduate Group, University of California, Berkeley, Berkeley, United States

**Abstract** Cytoplasmic dynein is a molecular motor responsible for minus-end-directed cargo transport along microtubules (MTs). Dynein motility has previously been studied on surface-immobilized MTs in vitro, which constrains the motors to move in two dimensions. In this study, we explored dynein motility in three dimensions using an MT bridge assay. We found that dynein moves in a helical trajectory around the MT, demonstrating that it generates torque during cargo transport. Unlike other cytoskeletal motors that produce torque in a specific direction, dynein generates torque in either direction, resulting in bidirectional helical motility. Dynein has a net preference to move along a right-handed helical path, suggesting that the heads tend to bind to the closest tubulin binding site in the forward direction when taking sideways steps. This bidirectional helical motility may allow dynein to avoid roadblocks in dense cytoplasmic environments during cargo transport.

## Introduction

Cytoskeletal motors transport a wide variety of intracellular cargos by processively moving along linear tracks. These motors do not always follow a linear trajectory. Rather, they generate torque perpendicular to their direction of motion, resulting in a helical movement relative to the filament. Such helical movement was first observed in a filament gliding assay in which surface-immobilized *Tetrahymena* axonemal dynein motors rotated MTs about their principal axes while translocating them (*Vale and Toyoshima, 1988*). Helical movement of cargoes has subsequently been demonstrated for several members of the myosin and kinesin superfamilies (*Nishizaka et al., 1993*; *Nitzsche et al., 2008*; *Yajima et al., 2008*; *Bormuth et al., 2012*; *Brunnbauer et al., 2012*).

Cytoplasmic dynein is the primary MT minus-end directed motor responsible for diverse cellular processes in cargo transport, nuclear positioning, and cell division (*Roberts et al., 2013*). Despite its importance, key aspects of dynein's mechanism remain unclear, including whether or not the motor can produce torque. In contrast to kinesin-1, which follows a single protofilament track (*Ray et al., 1993*; *Yajima and Cross, 2005*), dynein takes frequent sideways steps (*Reck-Peterson et al., 2006*). However, the full trajectory of dynein motors in three dimensions (3D) remains unknown, because the motors are sterically prevented from moving around the circumference of surface-immobilized MTs.

## Results

To track dynein motility in 3D, we constructed MT 'bridges' (*Brunnbauer et al., 2012*) (*Figure 1A*), which allow unconstrained motion between the protofilaments. The bridges were constructed by suspending a fluorescently labeled MT between two large (2 μm diameter) polystyrene beads, immobilized on the surface. These beads were densely coated with a chimeric protein containing the dynein MT binding domain (MTBD) that stably binds to MTs (*Gibbons et al., 2005*; *Carter et al., 2008*). Multiple dynein motors were linked to a 0.5-μm diameter bead (referred to as cargo) and brought in proximity of the MT bridge with an optical trap. When the motors bound to the MT, the

**\*For correspondence:** yildiz@ berkeley.edu

**Competing interests:** The authors declare that no competing interests exist.

**Reviewing editor**: Anthony A Hyman, Max Planck Institute of Molecular Cell Biology and Genetics, Germany

**eLife digest** Cells rely on 'molecular motors' travelling along tracks called microtubules to move proteins and other cargoes between different parts of a cell. Dynein is a molecular motor that moves along the microtubules by taking "steps" towards the slowly growing end of these tracks.

The trajectories of dynein motors have been studied extensively using techniques that can follow their movements in two dimensions. However, some molecular motors can also rotate as they travel, creating a twisting force called a torque that causes the motor to spiral around the microtubule in a helix.

To assess the torque that dynein can generate and to better understand its movements in three dimensions, Can et al. used a length of microtubule to build a 'bridge' between two polystyrene beads. The dynein motors were made to carry a smaller polystyrene bead as cargo, and the movement of this smaller bead was tracked using a computer algorithm to interpret the motion recorded by a microscope.

Can et al. found that dynein moves in a helical trajectory around the microtubule, rather than travelling along it in a straight line. As it travels it can twist in one direction or the other, generating torque in either direction. This is unlike other types of molecular motor, which produce torque in just one direction. However, dynein prefers to rotate to the right, suggesting that with every step along a microtubule, it binds to the closest available binding site in the forward direction.

Why might it be useful for molecular motors to behave in this way? Can et al. propose that the ability to rotate in both directions may allow dynein to avoid roadblocks or other obstacles in the dense and busy cellular environment in which it has to operate.

cargo bead was released from the trap and its motion was recorded with bright-field microscopy (*Figure 1B*). The *xy* position of the cargo was determined by a two-dimensional Gaussian tracking algorithm. The *z* position was determined by changes in the intensity of the bead center (*Figure 1C–D*), allowing us to track bead movement in 3D.

We initially tested the robustness of our experimental geometry using human kinesin-1 motors, which follow a single protofilament (*Ray et al., 1993*; *Yajima and Cross, 2005*). Because the handedness of the MTs varies based on the number (11–14) of protofilament tracks (*Hyman et al., 1995*), we polymerized tubulin with a non-hydrolyzable GTP analog (GMP-CPP). 96% of GMP-CPP MTs is made of 14 protofilaments and has a left-handed supertwist with 6400 ± 1000 nm pitch (*Hyman et al., 1995*). The average velocity of kinesin-driven beads was 541 ± 23 nm/s (mean ± SEM, N = 10), comparable to the speed of single motors (*Yildiz et al., 2008*). We observed that kinesin-1-coated beads travelled along GMP-CPP MTs with a left-handed helical motion with a pitch of 6500 ± 400 nm (mean ±SEM, N = 6) (*Figure 1—figure supplement 1*, *Video 1*), similar to the helicity of 14 protofilament MTs. The results are also consistent with the reported values of kinesin-1 movement on GMP-CPP MTs (5700–7900 nm pitch, left-handed) using alternative geometries (*Nitzsche et al., 2008*; *Brunnbauer et al., 2012*).

## Helical motility of dynein dimers

We next investigated the helical motility of cytoplasmic dynein. We used a tail-truncated yeast dynein artificially dimerized with glutathione S-transferase (GST-Dyn$_{331kD}$), which has similar motile properties to full-length dynein (*Reck-Peterson et al., 2006*). The motors were fused to GFP at the N-terminus and attached to cargo beads coated with anti-GFP antibodies (see 'Materials and methods'). At 1 mM ATP, dynein-coated beads moved in helical trajectories with a pitch of 591 ± 32 nm (mean ± SEM, 67 rotations, 15 beads, *Figure 2A–C*). This rotation corresponds to a sideways movement to a neighboring protofilament (6 nm) for every six tubulin dimers (48 nm) in the forward direction. Unlike axonemal dynein and kinesin motors, which primarily rotate along their tracks in only one direction (*Vale and Toyoshima, 1988*; *Brunnbauer et al., 2012*), beads coated with cytoplasmic dynein exhibited both left- and right-handed helical movement on MTs (N = 67 rotations, *Figure 2A–B*, *Figure 2—figure supplements 1 and 2*, *Videos 2,3*). The speeds of left- and right-handed movements along the helical path (79 ± 4.7 nm/s and 89 ± 17 nm/s, mean ± SEM, respectively) were statistically indistinguishable

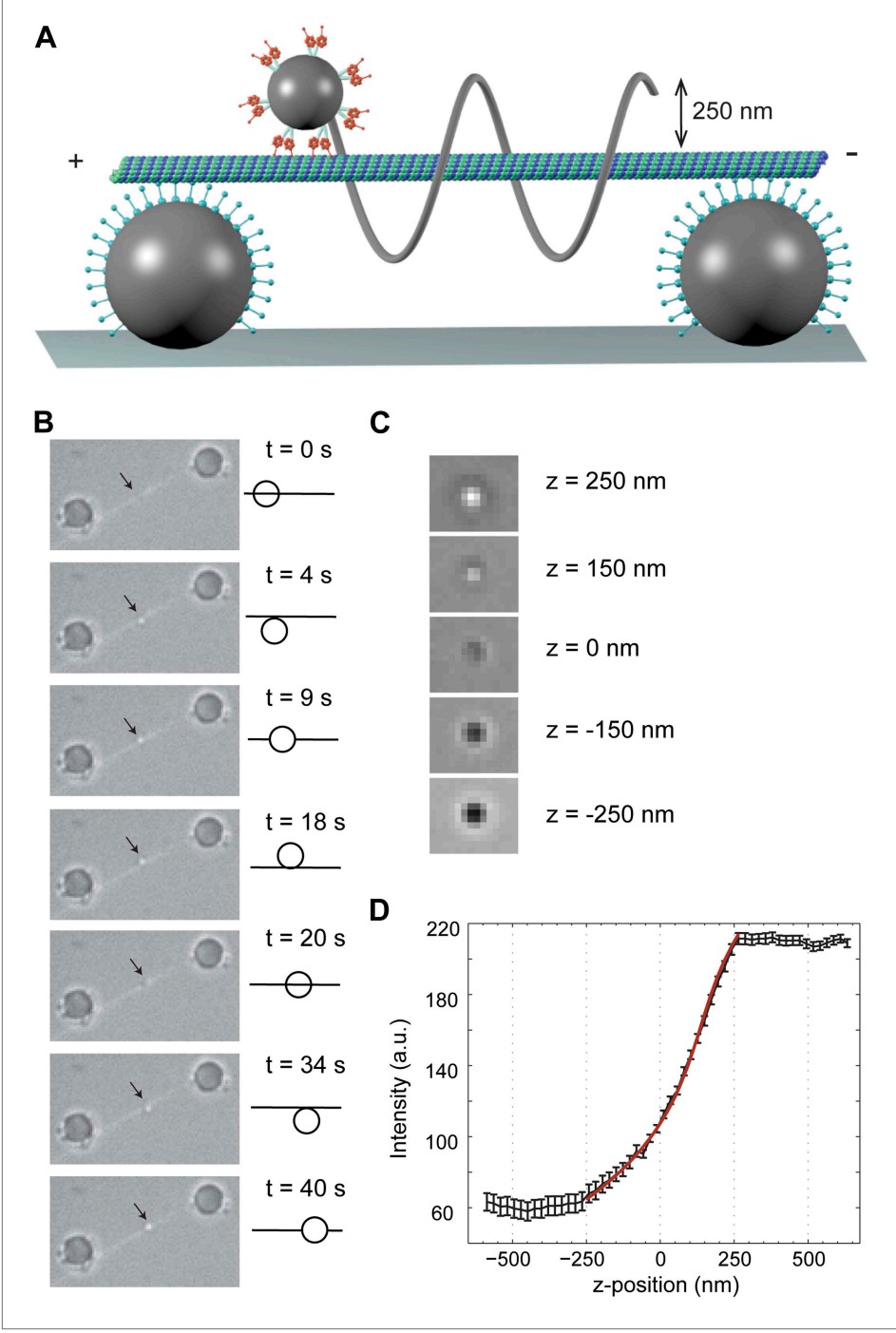

**Figure 1**. 3D tracking of dynein-driven transport along MT. (**A**) Schematic representation of the experimental geometry (not to scale). The MT suspended bridge is formed by attaching an MT (green) to the surface-immobilized beads (gray) that are coated with a chimeric protein containing the dynein MTBD. A 500 nm diameter cargo bead is coated with multiple dynein motors and trapped by a focused laser beam (not shown) for placement of the bead on the bridge. The bead center is expected to be separated by ~250 nm from the MT. (**B**) Movement of a GST-Dyn$_{331kD}$-coated bead along an MT bridge (left). The fluorescent image of the MTs has been superimposed onto the bright-field images. The bead moves in a left-handed helical manner along the MT. The schematic on the right represents the side view for the orientation of the bead relative to the MT. (**C**) Bright-field image of the cargo-bead in different z positions shows that the z position of the bead relative to MT can be determined by its brightness. Images are taken at z = −250 nm, −150 nm, 0 nm, +150 nm, +250 nm. (**D**) The averaged intensity of a
*Figure 1. Continued on next page*

*Figure 1. Continued*

500 nm diameter bead under a brightfield illumination at variable z positions. The averaged intensity from 20 beads increases as the bead is moved from −250 nm to +250 nm in the z direction relative to the image plane. The red curve represents a fit to a third order polynomial ($R^2 = 0.998$). The z position of a motor-coated bead was calculated from the calibration curve. Error bars represent SEM.

The following figure supplement is available for figure 1:

**Figure supplement 1**. Kinesin-1 follows a single protofilament on a MT track.

---

(t-test, p = 0.28). Bidirectional helical movement was also evident from traces of single beads, which occasionally switch direction during a run (4 out of 67 rotations, *Figure 2D*). In contrast, reversal of bead motility along the MT axis was never observed.

We next investigated the helical motility of a full-length yeast cytoplasmic dynein (*Reck-Peterson et al., 2006*). The motors were fused to GFP at the N-terminus and attached to cargo beads coated with anti-GFP antibodies. Dynein-coated beads moved in helical trajectories with a pitch of 500 ± 36 nm (mean ± SEM, *Figure 2—figure supplement 3*, *Video 4*) and exhibited both left- and right-handed helical movement. The velocities of left- and right-handed movement along the helical path (42 ± 11 nm/s and 43 ± 10 nm/s, mean ± SEM, respectively) were statistically indistinguishable (t-test, p = 0.21). Similar to GST-Dyn$_{331kD}$, beads driven by full-length dynein occasionally switch direction during a run (8 out of 33 rotations, *Figure 2—figure supplement 4*). Unlike GST-Dyn$_{331kD}$, which has a net preference for left-handed helical movement (75%), full-length dynein has a net preference for right-handed helical movement (58%, t-test, p = $10^{-5}$). This difference may be related to GST dimerization, in which the heads may be oriented differently relative to the MT surface.

The pitch of dynein-driven rotation is much shorter than the supertwist of the GMP-CPP MTs (~6400 nm), suggesting that helical motility of dynein is independent from the helicity of the MT track. To verify this, we repeated the assay with GST-Dyn$_{331kD}$ on taxol-stabilized MTs, which contain a mixture of 12 (77%) and 13 (11%) and 14 (2%) protofilaments (*Ray et al., 1993*). Out of 34 rotations, 68% were left-handed and 32% were right-handed. On average, the pitch of helical movement was 607 ± 50 nm (mean ±SEM) (*Figure 2—figure supplement 5*), similar to that of GMP-CPP MTs (t-test, p=0.77). The results demonstrate that our findings are not an artifact of the MT polymerization method.

To test the possibility whether bidirectional helical motility is driven by a rotational tug-of-war between dynein motors (i.e., some motors strictly rotate in a right-handed helix and the others rotate in the opposite direction), we tracked individual GST-Dyn$_{331kD}$ motors labeled with a quantum-dot on MT bridges (*Figure 3A*). The average run length of single motors on MT bridges was 1.5 ± 0.2 µm (mean ± SEM, N = 10), which is similar to that measured on surface-immobilized MTs (*Reck-Peterson et al., 2006*). The velocity of single dimers was 64 ± 8 nm/s (mean ± SEM, N = 10), similar (t-test, p = 0.12) to that of cargo beads carried by multiple dimers (80 ± 10 nm/s, N = 15). The traces of single motors show high variability along the perpendicular axis of MTs and switch directions in their sideways movement more frequently than the beads carried by multiple motors (*Figure 3B*). These results exclude rotational tug-of-war.

## Helical motility of dynein monomers

We next tested the possibility that the relative orientation of the two motor domains in a dimer can influence the handedness of helical motility. To eliminate the contribution of interhead orientation, we used a cargo bead coated with monomeric Dyn$_{331kD}$, which is not processive on its own (*Reck-Peterson et al., 2006*). We observed that multiple monomers were able to drive processive

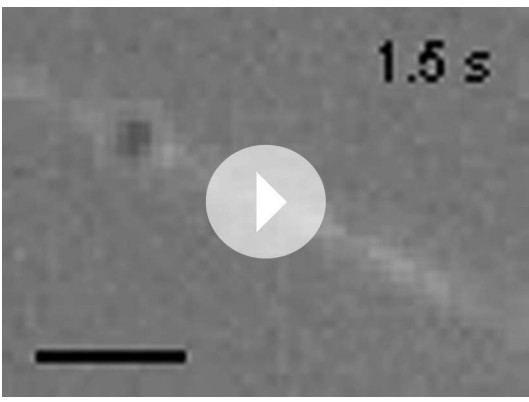

**Video 1**. Example recording of left-handed rotations of a kinesin-1 coated bead at 100 ms time resolution via bright-field microscopy. Fluorescent image of MT has been superimposed onto the bright-field image for illustration purposes.

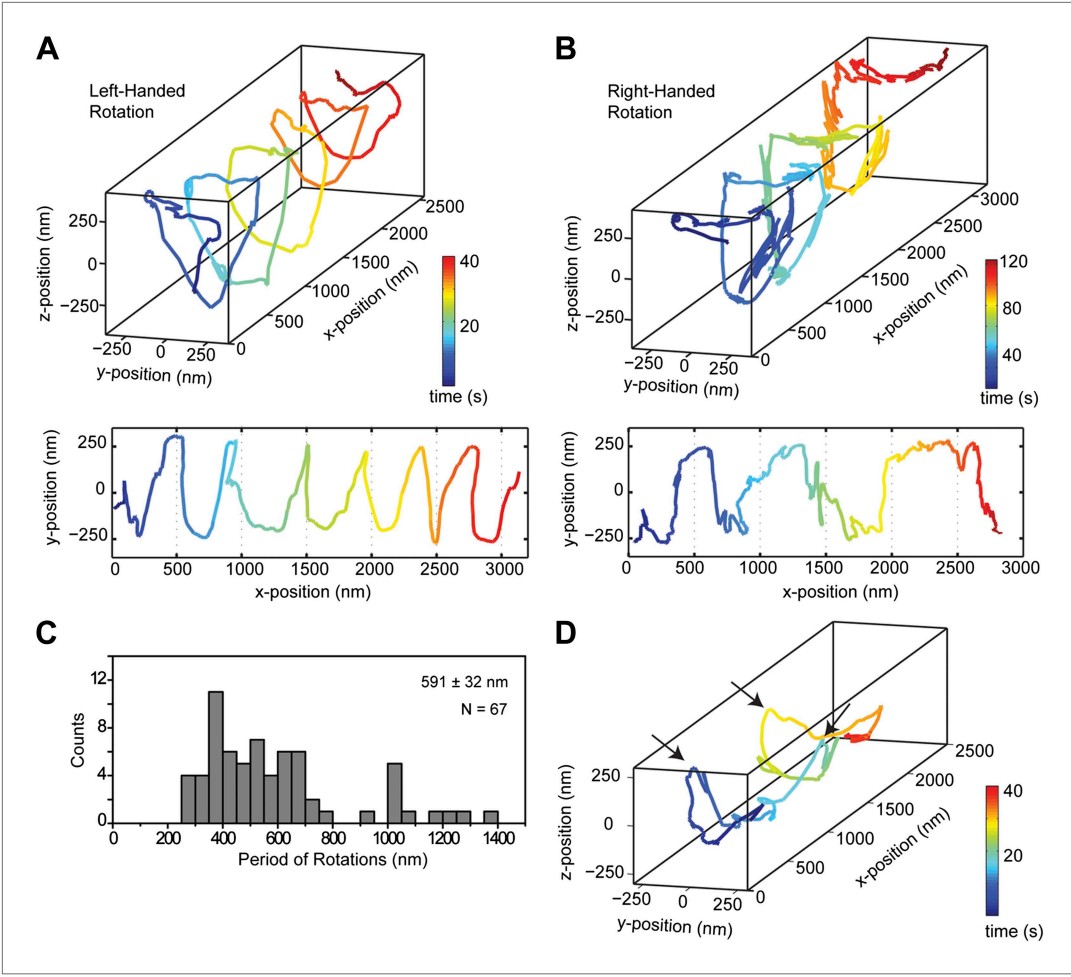

**Figure 2**. Dynein moves in both left- and right-handed helical paths along MT bridges. (**A** and **B**) (top) Representative three-dimensional trace of a cargo bead-driven by GST-Dyn$_{331kD}$ motors shows left- (**A**) and right-handed (**B**) helical motion. (bottom) Two-dimensional projections of the traces shown at top. (**C**) Histogram of observed pitches per complete rotation. The average pitch is 591 ± 32 nm (mean ± SEM). The average pitch of the left-handed movement (546 ± 42 nm, SEM, N = 32) was shorter (t-test, p = 0.01) than that of the right-handed movement (749 ± 81 nm, SEM, N = 10). (**D**) Change in handedness of rotation during the transport of a cargo bead. An example trace shows that a cargo bead initially moves along GMP-CPP MTs with a right-handed helical motion. At around t = 10 s, the bead reverses its helical motion for half of the period. At t = 20 s, the bead switches back to right-handed rotation and takes another half turn around the MT. Finally, at t = 25 s, the bead resumes left-handed helical motion until it disassociates from the MT. Arrows show the transitions from one type of helical motion to the other.

The following figure supplements are available for figure 2:

**Figure supplement 1**. Movement of a GST-Dyn$_{331kD}$ coated bead along an MT bridge.

**Figure supplement 2**. Additional example of the right-handed helical movement of a GST-Dyn$_{331kD}$-coated bead along an MT bridge.

**Figure supplement 3**. Bidirectional helical motility of cargo beads driven by full-length dynein along MT bridges.

**Figure supplement 4**. Change in handedness of rotation during the transport of a cargo bead driven by full-length dynein motors.

**Figure supplement 5**. GST-Dyn$_{331kD}$ on taxol stabilized MTs.

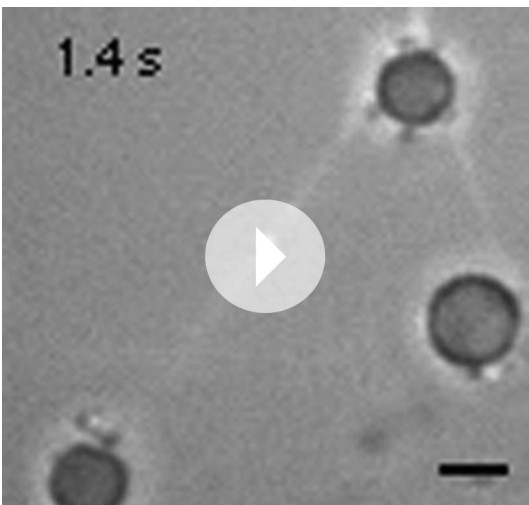

**Video 2**. Example recording of left-handed rotations of GST-Dyn$_{331kD}$-coated bead at 100 ms time resolution via bright-field microscopy.

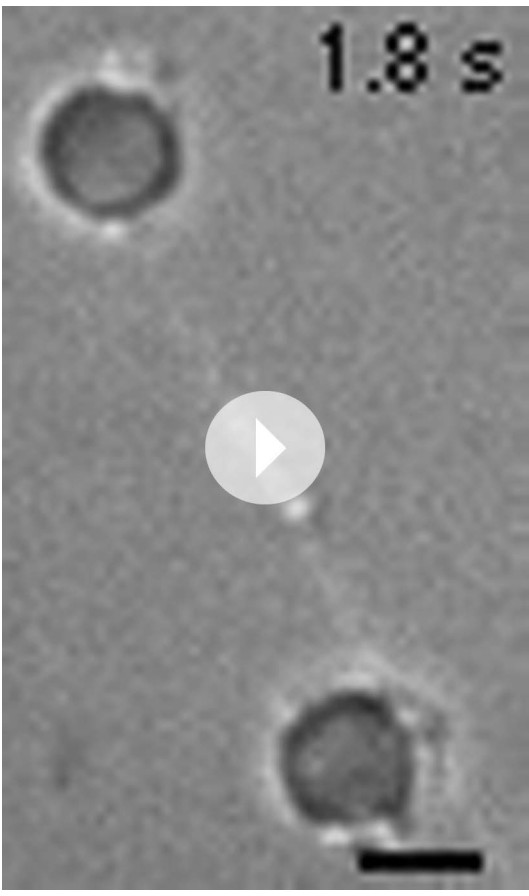

**Video 3**. Example recording of right-handed rotations of GST-Dyn$_{331kD}$-coated bead at 100 ms time resolution via bright-field microscopy.

motility of cargo beads at 61 ± 6 nm/s (mean ± SEM), consistent with the ability of multiple non-processive kinesins or myosins to drive processive motility (*Kamei et al., 2005*; *Furuta et al., 2013*).

The trajectories of beads driven by dynein monomers (*Figure 4*, *Video 5*) also displayed a helical component. The average pitch (579 nm ± 38 nm, 57 rotations, 11 beads) was similar to that of GST-Dyn$_{331kD}$ (t-test, p = 0.78) and higher than full-length dynein (t-test, p = 0.16). The majority (59%) of the rotations was right-handed, similar to full-length dimers (N-1 two proportion test, p = 0.47). The speeds of left- and right-handed movements along the helical path (66 ± 10 nm/s and 57 ± 6 nm/s, respectively) were statistically indistinguishable (t-test, p = 0.18). The results indicate that right-handed preference of full-length dynein for helical movement is not due to the head–head orientation of the dimer.

## Discussion

In this study, we showed that cytoplasmic dynein moves along MTs in a helical trajectory with a pitch of ~500 nm. Single dynein dimers take frequent sideways steps (*Reck-Peterson et al., 2006*), whereas multiple dynein dimers persistently move in a helical path with a net preference for a right-handed rotation. What is the molecular basis of this preference? Kinesin-1 monomers are believed to move to the closest possible site on the microtubule which stays to the left of the previous binding site (*Yajima and Cross, 2005*). However, kinesin-1 dimers only walk along a single protofilament because their short neck-linker prevents off-axis stepping (*Brunnbauer et al., 2012*). In dynein, monomers may prefer to attach to the nearest tubulin binding site towards the MT minus end, favoring a rightward step (*Figure 5A*). In the case of a dimer, the heads are attached to neighboring protofilaments and the leading head prefers to be on the right side at 30° relative to the trailing head (*DeWitt et al., 2012*; *Qiu et al., 2012*). This orientation would make a rightward step more favorable for the trailing head, resulting in a net rightward bias (*Figure 5B*).

In contrast to other motors studied to date, dynein moves along both left- and right-handed helical paths. The molecular basis of the switches in helical directionality remains unclear. The irregular stepping pattern of individual motors rules out the possibility of a rotational tug-of-war. Instead, we propose that the helical pattern of bead movement may be determined by the conformations of dynein motors associated with the MT track. It is likely that dyneins that are bound in

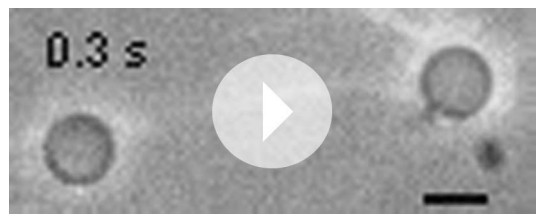

**Video 4.** Example recording of right- and left-handed rotations of full-length dynein-coated bead at 100 ms time resolution via bright-field microscopy.

**Figure 3.** Single dynein motors frequently switch the direction of their sideways movement. (**A**) Schematic representation of quantum-dot labeled single dynein motors on the MT bridges (not to scale). Expected amplitude of rotations is ~50 nm. (**B**) Two example traces show 2D projection of dynein motors along the MT, using fluorescent tracking. MT filaments remain nearly straight between the bridges (persistence length is 5.2 mm) and oscillate due to the thermal fluctuation. The red trace represents the fluctuation of the MT bridge in the perpendicular axis, determined by the position of a quantum dot stably bound to a MT. The red trace was subtracted from the traces of quantum dots attached to single dynein motors (blue trace). Single motors do not show signs of regular helical movement.

sufficiently close proximity to other dyneins experience steric exclusion effects. If a bead is carried by motors which are oriented such that one is bound immediately forward and to the right of the other, the tubulin binding sites ahead and to the right of the trailing head will be obstructed. Therefore, these motors may prefer to step to the left, and the entire cargo will eventually trace out a left-handed spiral (**Figure 5C**). The number and orientation of the motors that are simultaneously in contact with the MT may change over the course of a recording. Individual dynein motors have 1400 nm run length (**Reck-Peterson et al., 2006**), indicating that motors on the bead detach and reattach during a processive run of a cargo bead. These may alter the orientation of the MT bound motors, resulting in the reversal of helical directionality during a processive run (**Figure 5D**), as occasionally observed.

Inside cells, the MT surface is crowded with associated proteins, which act as roadblocks during the transport of intracellular cargos (**Stamer et al., 2002**). Furthermore, the intracellular space is crowded with large structures, such as vesicles and organelles. The ability of molecular motors to produce torque and axial force may allow the motors to switch protofilaments and avoid these obstacles. In cells, the same MT track is used for both plus- and minus-end-directed transport. Sideways movement may prevent traffic jams on MTs. In vitro assays have shown that dynein can bypass roadblocks whereas kinesin-1 stalls when it encounters an obstacle and eventually releases (**Dixit et al., 2008**). Bidirectional helical movement may provide additional flexibility to dynein to transport cargos in dense cellular environments.

## Materials and methods

### Protein preparation

*Saccharomyces cerevisiae* strains expressing mutant forms of cytoplasmic dynein (*Dyn1*) gene were generated by homologous recombination. Proteins were expressed and purified as described (**Reck-Peterson et al., 2006**).

MTs polymerized under 10 mM taxol contain a mixture of 12, 13, or 14 protofilaments. In MTs with 13 protofilaments, the protofilament long axes align with the MT long axis, whereas MTs with 12 and 14 protofilaments have a right-handed supertwist with 4000 nm pitch and left-handed supertwist with 6400 nm pitch, respectively (**Hyman et al., 1995**). GMP-CPP MTs were grown for 3 hr at 37°C from a 50 µl BRB80 buffer (80 mM PIPES pH 6.8 with KOH, 2 mM $MgCl_2$, 1 mM EDTA) solution supplemented with 5 µM tubulin (80% unlabeled porcine tubulin, 20% HiLyte 647 labeled porcine tubulin), 1 mM GMP-CPP and 2 mM $MgCl_2$. Assembled MTs were pelleted at 40,000×*g* with

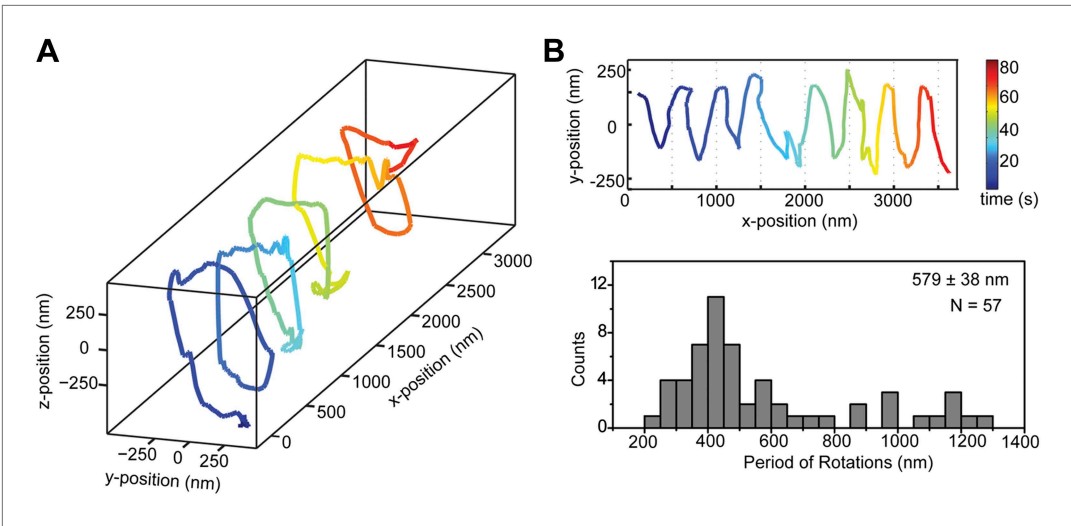

**Figure 4**. Dynein monomers prefer to move in a right-handed helix. (**A**) Representative three-dimensional trace of a cargo bead driven by monomers shows right-handed helical motion. (**B**) (top) Representative two-dimensional trace for monomeric Dyn$_{331kD}$. (bottom) Histogram of the periods of rotations shows that the average pitch is 579 ± 38 nm (mean ±SEM). The average pitch of the left-handed movement (658 ± 92 nm, SEM, N = 17) was longer (t-test, p=0.05) than that of the right-handed movement (490 ± 40 nm, SEM, N = 24).

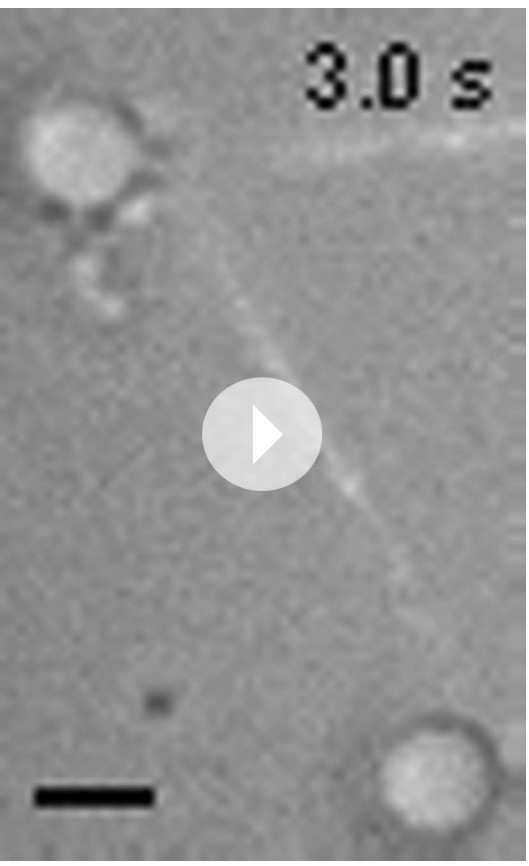

**Video 5**. Example recording of right-handed rotations of monomeric Dyn$_{331kD}$-coated bead. Video recorded via bright-field microscopy with time resolution of 100 ms.

Beckman Ti 102.1 rotor and resuspended in 60 µl BRB80 buffer. The average length of HiLyte 647 labeled MTs was 15 µm.

## Labeling beads with antibodies

Carboxylated polystyrene beads were coated with anti-rabbit polyclonal GFP antibodies (Covance, Emeryville CA). The beads were initially pelleted and resuspended in the activation buffer (100 mM MES, 100 mM NaCl, pH 6.0). Carboxyl groups on the surface of the bead were functionalized with amine reactive groups via EDC and sulfo-NHS crosslinking for 30 min at room temperature. The beads were then washed with phosphate buffer saline (PBS) at pH 7.4, and anti-GFP antibodies were added to the beads and reacted for 3 hr in room temperature. Excess antibodies were removed by centrifugation. The beads were resuspended in PBS along with 0.1% azide for storage purposes.

## Preparation of protein–bead complexes

eGFP was fused to the N-terminus of the SRS—dynein MTBD chimeric construct (GFP-SRS$_{85:82}$). GFP was used for attachment to an anti-GFP antibody-coated bead, and the MTBD stably binds to a MT (*Gibbons et al., 2005*; *Carter et al., 2008*). GFP-SRS$_{85:82}$ does not generate motility on its own (*Gibbons et al., 2005*; *Carter et al., 2008*). Saturating amount of GFP-SRS$_{85:82}$ was incubated with anti-GFP-coated carboxyl beads (2 µm diameter) on ice for 10 min in dynein assay

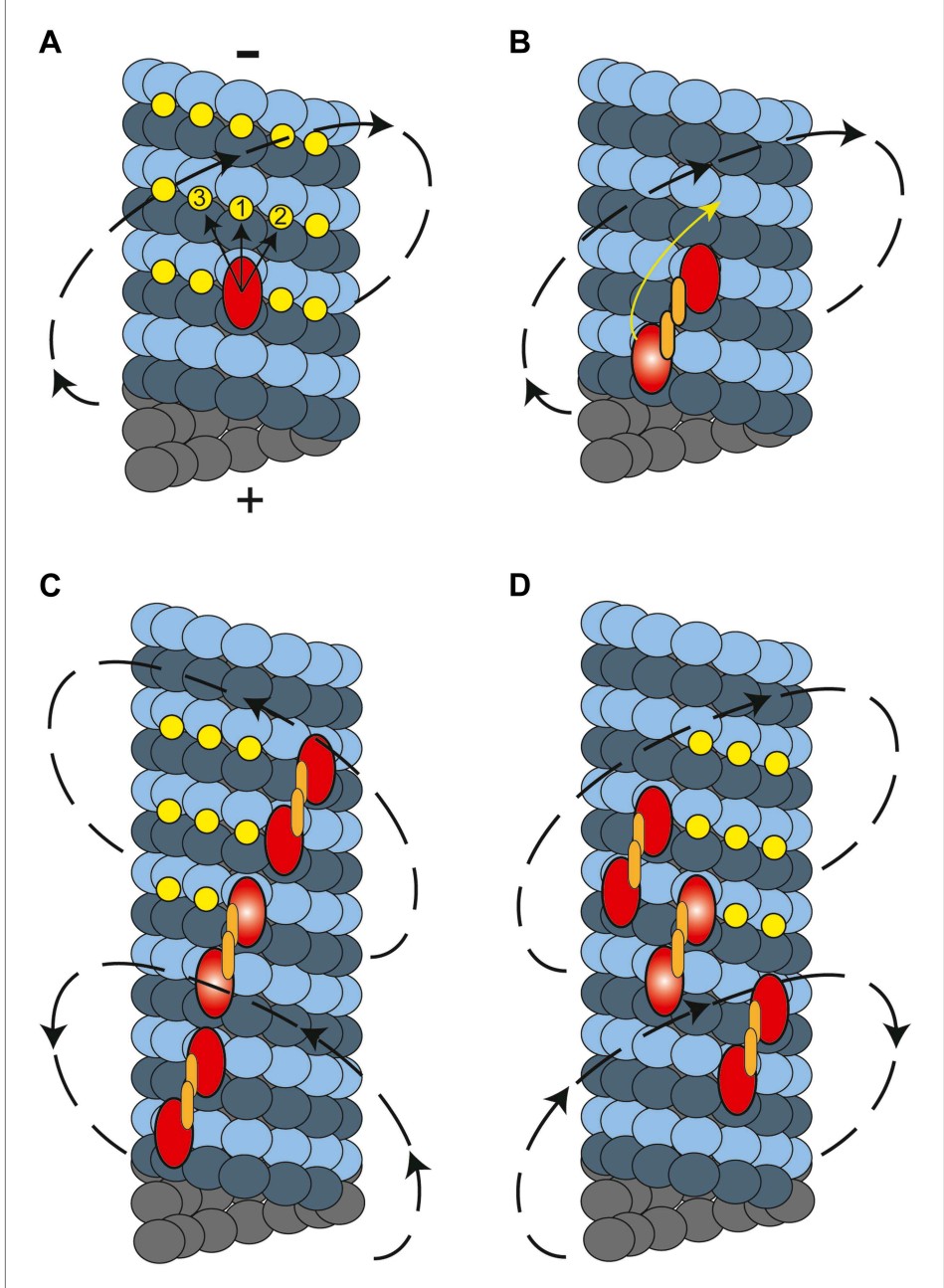

**Figure 5**. A model for the helical movement of cytoplasmic dynein. (**A**) Top view of a monomeric dynein (red oval) stepping toward the MT-minus end (arrows). The yellow circles represent the putative binding sites for the highlighted dyneins. The closest available binding sites are numbered from 1 to 3. The nearest (8 nm) binding site is along the same protofilament (1). The binding site on the right (2) has a shorter distance (9.3 nm) than the one in left (3, 10.8 nm), resulting in a net preference to step rightward. (**B**) A dynein dimer prefers to orient on an MT with the leading head positioned on the right of the trailing head. When the trailing head (bright red oval) moves forward, it prefers to step rightwards to be positioned on the right hand side of its partner. (**C**) When multiple dimers carry a cargo bead, helical directionality may be affected by the number and orientation of the motors associated with an MT track. In this orientation, tubulin binding sites to the right for the motor in the middle (bright red ovals) may be obstructed for the motor in the lead. This results in a tendency to move in a left-handed helical pattern. (**D**) Due to the finite run length of dynein motors, MT-associated motors dissociate and new ones attach to the track. Changes in the orientation of MT-bound motors, switch the directionality of helical movement.

buffer (DLB; 80 mM HEPES pH 7.4, 1 mM EGTA, 2 mM MgCl$_2$, and 10% glycerol) containing 1 mg/ml casein. Excess SRS protein was removed by centrifugation at 15,000×g, and the beads were resuspended in DLB.

## Preparation of flow chambers

SRS$_{85:82}$-coated beads were nonspecifically adsorbed to the coverslip. After 10 min of incubation, unbound beads were removed by washing the chamber twice with 30 µl DLB + 1 mg/ml casein. Casein was used to pre-block nonspecific surface attachment of MTs and motor proteins. Next, fluorescently labeled MTs were flowed and incubated for 10 min. Free MTs were washed with 30 µl DLB + 1 mg/ml casein buffer. We observed less than 20 bridges in 1000 µm × 1000 µm area. The bead density was kept high (four beads on average in 20 µm × 20 µm area), and MT concentration was kept low to ensure that each bridge was formed by a single MT. Motor-coated beads always moved unidirectional without changing the direction of motion along the MT long axis during processive motility, excluding the possibility of bridges containing multiple MTs pointing in the opposite directions. Finally, a solution containing dynein-coated 0.5-µm diameter beads in DLB buffer supplemented with 2.5 mM PCA (protocatechuic acid) and 50 nM PCD (protocatechuate-3,4-dioxygenase) oxygen scavenging system (*Aitken et al., 2008*), 1 mg/ml casein and 1 mM ATP were flowed to the chamber and sides of the chamber were sealed with nail polish to prevent evaporation of the assay solution.

## Data collection

The assays were performed with a custom-built optical trapping microscope equipped with Nikon TiE microscope body, Nikon 100× 1.49 NA plan apochromat objective and Ixon+ electron multiplied charge coupled device (EM-CCD) camera (Andor, United Kingdom). HiLyte-labeled MTs were excited with 632 nm laser beam in epifluorescence mode, and the fluorescent signal was detected by the EM-CCD camera with an effective pixel size of 160 nm. The videos were recorded at 100 ms frame rate. The surface of the flow chamber was scanned to find MT bridges between the two 2-µm-diameter beads. We performed our bead tracking assays on the bridges, in which MTs are 10–15 µm long between the beads, appear steady by the resolution of a fluorescence microscope (250 nm) at 10 Hz frame rate, and the entire MT fluorescence appears in focus.

A 0.5-µm cargo bead freely diffusing in solution was trapped by a focused 1064 nm laser beam. The trap was steered with a pair of acousto-optical deflectors (AA Opto-Electronic, France), and bead position was detected by a position sensitive detector using back–focal plane interferometry. Leakage of the intense trapping beam to fluorescence detection was blocked by a 708/75 nm bandpass filter.

A single MT bridge was used over the course of one experiment. The dynein-coated beads were captured with an optical trap and brought to the proximity of the MT. When the motors on the cargo bead attached to the MT and started to move, the optical trap was turned off. The movement of beads was recorded using bright-field microscopy. MT polarity was first tested by placing the bead on a MT at the center of the bridge. Once the directionality is determined, the bead was moved away from the MT and placed at the plus-end tip of the bridge to explore the motility throughout the entire length of the bridge. All of the beads moved towards the same direction on a single bridge, without any reversals in axial direction.

The movement of single quantum dots on MT bridges was determined by labeling GST-Dyn$_{331kD}$ motors at the C-terminus with a quantum dot 655 using HaloTag attachment (*DeWitt et al., 2012*). Because the amplitude of side to side movements was smaller in the case of Q-dots (~50 nm) compared to beads (~250 nm), the MT fluctuation was subtracted from Quantum-dot labelled dynein data. To measure the oscillation of the MT bridges, the MTs were sparsely labeled with a quantum dot 585. The standard deviation of the position of 585 quantum dots was 35 nm in perpendicular direction and 17 nm in parallel direction to the long axis of the MTs. The position of 585 quantum dots was subtracted from the quantum dot 655 labeled dyneins to correct for MT oscillations.

## Data analysis

Cargo beads were tracked by using custom-written software in Matlab (The Mathworks, Natick MA), which utilizes Gaussian fitting to determine the *xy* position of the bead. The Matlab code used in data analysis is available at http://research.physics.berkeley.edu/yildiz/SubPages/code_repository.html and in *Supplementary file 1*. The precision of bead tracking was 3 nm in *x* and 5 nm in *y* directions. The *z* position of the bead was determined by the intensity of the bead center. To calibrate the

bead intensity as a function of $z$ position, the surface of a sample chamber was decorated with 0.5-μm diameter beads. The microscope objective was moved ±250 nm in the $z$ direction with 25 nm increments using PIFOC objective scanner (Physik Instrumente, Germany). Corresponding intensities of peaks at each frame were plotted with the $z$ position relative to the microscope objective. This routine was repeated for 20 times to obtain a calibration curve shown in *Figure 1D*. The bead intensity profile was fitted with a third order polynomial function and the $z$ position of the bead was calculated from the calibration curve. The bead image could not be well-fit by a Gaussian when the $z$ position was between −25 nm and +25 nm. To avoid sample-to-sample variability in calibration procedures, the background was subtracted and the intensities were normalized in the bright-field images of the bead. Traces of bead motility were smoothed by the moving average filter of window of five data points.

The pitches of the helical motion of the beads are corrected for the 6400 nm left-handed supertwist of GMP-CPP MTs. The corrected pitch is calculated by $\frac{1}{pitch_{measured}} = \pm\frac{1}{pitch_{MT}} + \frac{1}{pitch_{corrected}}$. The corrected pitch is longer than the measured pitch for left-handed rotations and shorter for right-handed rotations.

## Acknowledgements

We thank FB Cleary, V Belyy, and other members of the Yildiz laboratory for helpful discussions, and SC Howes and E Nogales for help on MT preparations. This work has been supported by NIH (GM094522 (AY)), NSF CAREER Award (MCB-1055017 (AY)), and Burroughs Welcome Foundation (AY).

## Additional information

### Funding

| Funder | Grant reference number | Author |
| --- | --- | --- |
| National Institutes of Health | GM094522 | Ahmet Yildiz |
| National Science Foundation | MCB-1055017 | Ahmet Yildiz |
| Burroughs Wellcome Fund | 1006579 | Ahmet Yildiz |

The funders had no role in study design, data collection and interpretation, or the decision to submit the work for publication.

### Author contributions

SC, Conception and design, Acquisition of data, Analysis and interpretation of data, Drafting or revising the article; MAD, Conception and design, Analysis and interpretation of data; AY, Conception and design, Drafting or revising the article

## Additional files

### Supplementary file

• Supplementary file 1. Matlab code used in data analysis.

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
