## [Decision Letter]

Thank you for sending your work entitled “Bidirectional helical motion of cytoplasmic dynein around microtubules” for consideration at *eLife*. Your article has been favorably evaluated by Tony Hunter (Senior editor) and 3 reviewers, one of whom is a member of our Board of Reviewing Editors, and one of whom, Andrew Carter, has agreed to reveal his identity.

The Reviewing editor and the other reviewers discussed their comments before we reached this decision, and the Reviewing editor has assembled the following comments to help you prepare a revised submission.

In this study, the authors investigate the 3 dimensional motion of cytoplasmic dynein as it walks along a microtubule. In the cell, the nature of this movement has important implications for how cargo is transported and how motors navigate the crowded cytoplasmic environment. Kinesin-1 precisely follows a single protofilament track on a microtubule, while dynein has been shown to take frequent sideways steps to neighboring protofilaments. However, since most work has been done on surface-immobilized MTs, which constrains movement to two dimensions, little is known about dynein's 3-dimensional movement and whether it can generate torque. The surprise from this work by Can et al. is that the dynein bead shows long distance helical motion in both directions.

We would like in principle to publish your paper in *eLife.* However there are two sets of concerns that we would like addressed.

First, we feel that there needs to be more discussion of your work. You should include a discussion on the importance of the ability of dynein to generate torque in the cell. You bring up the importance of torque in the Introduction, but do not mention anything about it in the Discussion. A detailed discussion on the molecular basis of bi-directionality is called for. Since bidirectional helical motion is the major finding of the paper, this warrants a more thorough explanation of the mechanism. How could orientation of the motors lead to changes in bead step? How could the number of motors affect this motion? What, specifically, could lead monomeric and dimeric dynein to behave differently? A model figure would be very useful to explain this.

Second, there are a number of points that need to be tightened up before publication, with respect to the presentation of the data, and some of its implications. This is a list of points from one of the reviewers, with which I agree. Please consider these when revising your manuscript.

1) Most of the experiments are done with yeast dynein artificial dimers, which are reported to take a large number of processive steps before detaching from a microtubule (Reck-Peterson 2006). The authors have data both on the processive movement of beads powered by multiple yeast dynein artificial dimers and on the processive movement of beads powered by single yeast dynein artificial dimers. Antibodies are used to capture this dynein on to the beads and this defines the bead attachment geometry. Clarifying this at the outset is really important because the authors show that the behavior of multi-dynein dimer beads differs from that of single dynein dimer Qdots (Figure 2—figure supplement 1). I tend to think that these data belongs in the main Figure 2. The heading 'Helical movement of dynein dimer' should at the very least be dimerS, since the authors describe helical movement of beads carrying multiple dynein artificial dimers.

2) Kinesin-1 is used as a positive control to demonstrate that tracking of the long-pitch helical path of protofilaments in 14PF GMPCPP MTs is accurately reported. It would be helpful to know if beads in such experiments always moved unidirectionally along MT bridges, or whether some MT bridges contained multiple MTs with some potentially pointing in the opposite direction to others. The fluorescence signal from the MTs should also help confirm that only single MTs are in the bridges. If the kinesin velocities showed a narrow Gaussian distribution that can also usefully inform the dynein experiments.

3) Assuming [2] above does indicate 1 MT per bridge, in experiments detecting bidirectional rotational motion it is important to compare the velocities of dynein-driven bead motility in the two directions, and not just the helical pitches. In Figure 2, “speeds of left handed and right handed movements were statistically indistinguishable”;does this refer to speeds in the X direction only or along the helical paths? The pitches of the helical paths are statistically different. It is also really important to clarify that in this 'bidirectional' motion, the direction of net progress along the MT axis is *not* changing: both left handed and right handed helical paths have the motor heading in the same direction along the MT axis.

4) Regarding the brightfield method for tracking the Z position of the beads, I have not seen brightfield used to do this before, and have slight concerns because the bright-dark appearance is highly nonlinear with Z (Figure 1) so that the bead goes briefly invisible as it flips between black and white quite suddenly in the videos and then remains black or white until its next flip. This needs more careful explanation, considering it looks like the brightfield signal may be insensitive to Z in parts of the trajectories. Could the flat tops of the trajectories in Figure 2 reflect this? The side to side (Y) tracking should not have this problem. Please clarify that the side-to-side tracking is using centroids and that both black and white images are included. Figure 2 does not currently make this clear. It might be better to plot X versus time and Y versus time and Z versus time one underneath the other for each sequence. This would be especially useful for Figure 2, where it is hard to see the change in direction / handedness. The bw signal from the bead could be coded into these plots.

5) Figure 2—figure supplement 1: 'The oscillation of the MT bridges': please clarify what this is. Is the MT bridge moving side to side as the Qdot squeezes underneath or is this just random positional noise of the MT bridge? “Expected amplitude of rotation is 60nm”: Please clarify, why? Is this single dynein dimer in fact rotating around the MT or is it just drifting randomly side to side? What is the observed run length distribution? In the example, a single motor moves more than 2 µm, implying >250 8 nm steps. The velocity of multi-dynein dimer driven beads is 61 nm sec-1 (clarify that this is along the MT axis), what is the velocity of these single dimer Qdots?

6) Finally there is a concern that cytoplasmic dynein in vivo is not thought to rotate its cargo around the MT axis. The experiment reported here clearly shows that dynein monomers have an intrinsic tendency to step off-axis, and that yeast artificial dynein dimers retain this tendency. I was left wondering whether authentic cytoplasmic dynein dimers might be better at tracking the MT axis, just as kinesin-1 dimers can over-ride the intrinsic tendency of kinesin monomers to drift off axis. The discussion concludes firmly that (yeast) dynein rotates its cargoes around the MT with a pitch of about 600nm. It is worth pointing out that blocking this rotation, as it is implied happens when MTs are bound to the coverslip surface, does not impede axial motion.

---

## [Author Response]

*First, we feel that there needs to be more discussion of your work. You should include a discussion on the importance of the ability of dynein to generate torque in the cell. You bring up the importance of torque in the Introduction, but do not mention anything about it in the Discussion*.

We added a new paragraph, starting “Inside cells, the MT surface is crowded with associated proteins, which act as roadblocks during the transport of intracellular cargos (17)...”

A detailed discussion on the molecular basis of bi-directionality is called for. Since bidirectional helical motion is the major finding of the paper, this warrants a more thorough explanation of the mechanism. How could orientation of the motors lead to changes in bead step? How could the number of motors affect this motion?

We added a paragraph starting “In contrast to the other motors studied to date, dynein can move along both left- and right-handed helical paths. The molecular basis of the switches in helical directionality remains unclear…”

What, specifically, could lead monomeric and dimeric dynein to behave differently?

In this revised manuscript, we tested whether the differences between monomeric and dimeric dyneins could be due to the GST-dimerization, which may orient the heads differently. To test this idea, we repeated the MT bridge assay using full-length yeast cytoplasmic dynein. We observed that full-length dynein also shows bidirectional rotation and reversals in helical directionality during a processive run of a bead, similar to GST-truncated dynein. Unlike GST-dimerized truncated dynein, full-length dyneins prefer to walk along a right-handed helical path. As a result, we removed the statement that monomeric and dimeric dyneins behave differently. We also added the following paragraph to the Discussion, starting “In this study, we showed that cytoplasmic dynein moves along MTs in a helical trajectory with a pitch of ∼500 nm. Single dynein dimers move along microtubules by taking frequent sideways steps (15), whereas multiple dynein dimers persistently move in a helical pattern with a net preference for a right-handed rotation…”

*A model figure would be very useful to explain this*.

We have added the paragraphs above to discussion and added a new model figure, Figure 5, to the revised manuscript.

*Second, there are a number of points that need to be tightened up before publication, with respect to the presentation of the data, and some of its implications. This is a list of points from one of the reviewers, with which I agree. Please consider these when revising your manuscript*.

1) Most of the experiments are done with yeast dynein artificial dimers, which are reported to take a large number of processive steps before detaching from a microtubule (Reck-Peterson 2006). The authors have data both on the processive movement of beads powered by multiple yeast dynein artificial dimers and on the processive movement of beads powered by single yeast dynein artificial dimers. Antibodies are used to capture this dynein on to the beads and this defines the bead attachment geometry. Clarifying this at the outset is really important…

We modified the corresponding section to:

“We next investigated the helical motion of a tail-truncated yeast cytoplasmic dynein artificially dimerized with glutathione S-transferase (GST-Dyn_331kD_), which has similar motile properties to full-length dynein (15). The motors were tagged with GFP at the N-terminus and attached to cargo beads coated with anti-GFP antibody (see Methods).”

*The authors show that the behavior of multi-dynein dimer beads differs from that of single dynein dimer Qdots (*Figure 2—figure supplement 1*). I tend to think that these data belongs in the main*
Figure 2.

We were not able to fit these data to Figure 2 (all panels and the legend have to fit into a single page). Instead, we put it as new Figure 3.

*The heading 'Helical movement of dynein dimer' should at the very least be dimerS, since the authors describe helical movement of beads carrying multiple dynein artificial dimers*.

This is fixed.

*2) Kinesin-1 is used as a positive control to demonstrate that tracking of the long-pitch helical path of protofilaments in 14PF GMPCPP MTs is accurately reported. It would be helpful to know if beads in such experiments always moved unidirectionally along MT bridges, or whether some MT bridges contained multiple MTs with some potentially pointing in the opposite direction to others. The fluorescence signal from the MTs should also help confirm that only single MTs are in the bridges*.

Motor-coated beads always moved unidirectionally along the microtubules, without changing the direction. This excludes the possibility of bridges containing multiple MTs pointing in opposite directions. We kept the bead concentration high and the MT concentration low such that there are many beads sticking to the surface, but very few bridges forming on the entire coverslip surface. Therefore, it is highly unlikely that multiple MTs form single bridges. We also carefully examined each bridge by fluorescence in order to ensure that the bridges are straight, and the bead pair contains a single MT bridge in between them. The average distance between the beads on the sample is around 15 µm, and there are 4 beads on average observed in 20 µm x 20 µm area. At this density of beads, we observe less than 20 bridges in 10^6^ µm^2^.

Finally, we use the same bridge for testing the movement of many beads. MT polarity is first tested by placing the bead on a MT at the center of the bridge. Once the directionality is determined, the bead is moved away from the MT and placed at the plus end tip of the bridge to explore the motility throughout the entire length of the bridge. ALL of the beads moved towards the same direction on a single bridge, without any reversals.

These statements are added to the Methods.

*If the kinesin velocities showed a narrow Gaussian distribution that can also usefully inform the dynein experiments*.

The average velocity of kinesin molecules is 541 ± 73 nm/s (mean ± SD, N=10). This comparable to the speed of single motors of the same truncated version of human kinesin-1 (hK560-GFP)(Yildiz et al., Cell 2008). This is added to the text.

*3) Assuming [2] above does indicate 1 MT per bridge, in experiments detecting bidirectional rotational motion it is important to compare the velocities of dynein-driven bead motility in the two directions, and not just the helical pitches. In*
Figure 2*, “speeds of left handed and right handed movements were statistically indistinguishable”;does this refer to speeds in the X direction only or along the helical paths? The pitches of the helical paths are statistically different*.

The speeds of the beads are calculated along the helical pitch.

Pitch corrected velocities for dimers (mean ± SEM):

Left Handed Rotations: 79 ± 4.7 nm/s

Right Handed Rotations: 89 ± 17 nm/s

One tail t-test between left and right velocities p=0.28 (statistically indistinguishable)

Pitch corrected velocities for monomers (mean ± SEM):

Left Handed Rotations: 66 ± 10 nm/s

Right Handed Rotations: 57 ± 6 nm/s

One tail t-test between left and right velocities p=0.18 (statistically indistinguishable)

One tail t-test between dimer vs monomer p=0.004 (statistically distinguishable)

These numbers are corrected in text and the issue is now clarified.

*It is also really important to clarify that in this 'bidirectional' motion, the direction of net progress along the MT axis is* not *changing: both left handed and right handed helical paths have the motor heading in the same direction along the MT axis*.

Both left and right handed rotating dynein molecules moved in a unidirectional manner. While we observed change in the helicity of movement during the processive runs of the beads, we never observed a change in the directionality along the MT long axis. We clarified this in the text.

4) Regarding the brightfield method for tracking the Z position of the beads, I have not seen brightfield used to do this before…

Brightfield imaging has been used to determine the Z position of the bead in previous studies. Most recently, the Okten lab used this method to track kinesin-1-driven beads in three dimensions ([3], cited in our manuscript).

*I have slight concerns because the bright-dark appearance is highly nonlinear with Z (*Figure 1*)…*

The bright-dark appearance of the bead is nearly linear ± 150 nm and becomes highly nonlinear between 150 and 250 nm. We fit our calibration curve to a third order polynomial to convert bead signal to the z position (R2=0.9982). The calibration curve (Figure 1, updated) and description for how we extrapolated the z position from this curve are added to the manuscript.

*The bead goes briefly invisible as it flips between black and white quite suddenly in the videos and then remains black or white until its next flip. This needs more careful explanation*.

The bead never becomes fully invisible to the eye. At z = 0 position, the bead does not become invisible (see Figure 1), instead it contains a dark ring near its center. In our imaging conditions, we cannot reliably detect the bead center when the z position is between -25 nm and +25 nm. This is now added to the methods. However, this does not affect our ability to observe rotation of the beads around MTs.

*It looks like the brightfield signal may be insensitive to Z in parts of the trajectories. Could the flat tops of the trajectories in*
Figure 2
*reflect this? The side to side (Y) tracking should not have this problem*.

The reviewers are correct. As we explain in Figure 1, the brightfield signal is sensitive to z within ±250 nm of the focal plane. If the bead moves higher or lower than this limit, we accept these positions as +250 nm and -250 nm, respectively. Traces in Figure 2&B appear flat at the top of the helical turn, because the bead is beyond the detectable range in the z-axis. The primary purpose of the z measurement is to determine the handedness of the helical trajectory, and this is still clear from the data presented in Figure 2.

*Please clarify that the side-to-side tracking is using centroids and that both black and white images are included.*
Figure 2
*does not currently make this clear*.

Side to side tracking is done by using 2D Gaussian algorithm (plus a constant term for the background) and the both white and black images are included. After beads are tracked in 2 dimensions, the intensities of corresponding center positions are obtained from original recordings for Z tracking. We have also performed centroid calculations to verify that our conclusions are not an artefact of the Gaussian fitting algorithm. The figure below shows that centroid and Gaussian algorithms have slight disagreements in the *xy* position, but the rotational pitch of the bid does not change (Figure 6).Author response image 1.

*It might be better to plot X versus time and Y versus time and Z versus time one underneath the other for each sequence. This would be especially useful for*
Figure 2*, where it is hard to see the change in direction / handedness. The bw signal from the bead could be coded into these plots*.

In the revised version, we show example traces with x, y and z plotted separately as a function of time, and encoded the bead image in Figure 2—figure supplement 2.

*5)*
Figure 2—figure supplement 1*: 'The oscillation of the MT bridges': please clarify what this is. Is the MT bridge moving side to side as the Qdot squeezes underneath or is this just random positional noise of the MT bridge?*

The persistence length of MT’s is 5.2 mm (Gittes et al., 1993) which is much longer than the length of our MT bridges. Therefore, MTs remain straight between the bridges and oscillate due to thermal noise. To measure amplitude of this random oscillation, MTs were sparsely labeled with a quantum dot 585 using bition-streptevidin linkage. The standard deviation of the quantum dot position was 35 nm in perpendicular direction and 17 nm in parallel direction to the long axis of the MTs. These errors are much higher than the localization error from the Gaussian tracking. Therefore, we interpreted that large fluctuations of fixed positions at MT bridges are due to the thermal fluctuations. We included a statement to this effect in the figure legend.

“Expected amplitude of rotation is 60nm”: Please clarify, why? Is this single dynein dimer in fact rotating around the MT or is it just drifting randomly side to side?

We modified this distance to 50 nm. The distance corresponds to the estimated amplitude of rotation for a quantum-dot labeled dynein dimer rotating around the MT. Given the MT radius (12 nm), the estimated distance between dynein MTBD and the GST (∼25 nm) and the radius of the quantum dots (∼12 nm), the expected radius of rotation is ∼50 nm. Compared to the beads carried by multiple dynein motors, single motors show higher variability and more frequent switches in sideways movement. So the single dynein dimer is not rotating as uniformly as cargo bead carried by multiple dyneins around the MT.

What is the observed run length distribution? In the example, a single motor moves more than 2 µm, implying >250 8 nm steps. The velocity of multi-dynein dimer driven beads is 61 nm sec-1 (clarify that this is along the MT axis), what is the velocity of these single dimer Qdots?

61 nm/s is the speed of the monomers. We fixed this error.

The run lengths of single dynein motors on MT bridges is 1.5 ± 0.7 µm (mean ± SD, N= 10), which is similar to that measured on surface-immobilized MTs (Reck-Peterson et al. *Cell* 2006). Velocity of single dynein dimers is 64 ± 8 nm/s (mean ± SEM , N = 10), similar (t-test p= 0.12) to that of cargo beads carried by multiple dynein motors (80 ± 13 nm/s, N = 15).

*6) Finally there is a concern that cytoplasmic dynein in vivo is not thought to rotate its cargo around the MT axis. The experiment reported here clearly shows that dynein monomers have an intrinsic tendency to step off-axis, and that yeast artificial dynein dimers retain this tendency. I was left wondering whether authentic cytoplasmic dynein dimers might be better at tracking the MT axis, just as kinesin-1 dimers can over-ride the intrinsic tendency of kinesin monomers to drift off axis. The discussion concludes firmly that (yeast) dynein rotates its cargoes around the MT with a pitch of about 600nm. It is worth pointing out that blocking this rotation, as it is implied happens when MTs are bound to the coverslip surface, does not impede axial motion*.

Although the reviewers did not strictly require this experiment, we tested the motility of cargo beads driven by full length yeast cytoplasmic dynein. We added the following paragraph, new Video 4 and two supplemental figures to the revised manuscript. The reason why we did not place this data in the main text is because we are not able to express the full-length protein at a high yield. When overexpressed, protein aggregation sharply reduces yield and protein quality. Under a native promoter, the protein behaves well, but the yield is comparatively poor, making it difficult to efficiently coat the cargo beads with dynein. As a result the 3D traces do not appear as high quality as that of GST-Dyn_331kDa_, because the smoothness of rotations requires more than a few motors on a bead. If the reviewer panel feels that the full length data need to be moved to the main text, we would be happy to do so.

“We next investigated the helical motility of a full-length yeast cytoplasmic dynein (15)*.* The motors were fused to GFP at the N-terminus and attached to cargo beads coated with anti-GFP antibodies. Dynein-coated beads moved in helical trajectories with a pitch of 500 ± 36 nm (mean ± SEM, Figure 2—figure supplement 3, Video 4) and exhibited both left- and right-handed helical movement. The speeds of left- and right-handed movements along the helical path (42 ± 11 nm/s and 43 ± 10 nm/s, mean ± SEM, respectively) were statistically indistinguishable (t-test, p = 0.21). Similar to GST-Dyn_331kD_, beads driven by full-length dynein occasionally switch direction during a run (8 out of 33 rotations, Figure 2—figure supplement 4). Unlike GST-Dyn_331kD_, which has a net preference for left-handed helical movement (75%), full-length dynein has a net preference for right-handed helical movement (58%, t-test, p = 10^-5^). This difference may be related to GST dimerization, in which the heads are oriented differently relative to the MT surface.”